# Upper Extremity Skeletal Muscle Mass Asymmetry Exacerbated by Shoulder Imbalance in Lenke1A Adolescent Idiopathic Scoliosis

**DOI:** 10.3390/jcm11237117

**Published:** 2022-11-30

**Authors:** Tetsuro Ohba, Go Goto, Nobuki Tanaka, Kotaro Oda, Marina Katsu, Hayato Takei, Kensuke Koyama, Hiroki Oba, Hirotaka Haro

**Affiliations:** 1Department of Orthopaedic Surgery, University of Yamanashi, Yamanashi 4008510, Japan; 2Department of Orthopaedic Surgery, Shinshu University School of Medicine, Nagano 3908621, Japan

**Keywords:** adolescent idiopathic scoliosis, Lenke1A curves, Lenke5C curves, shoulder imbalance, skeletal muscle asymmetry

## Abstract

Limb muscle strength asymmetry affects many physical abilities. The present study (1) quantified limb muscle asymmetry in patients with adolescent idiopathic scoliosis (AIS); (2) compared AIS patients with major thoracolumbar/lumbar (TL/L) or major thoracic (MT) curves; (3) examined correlations between limb muscle asymmetry and radiographic parameters. Patients with AIS with major TL/L curves (Lenke type 5C) and MT curves (Lenke Type 1A) who underwent posterior spinal fusion at our university hospitals were included. Patients with left hand dominance were excluded. Body composition was measured using whole-body dual-energy X-ray absorptiometry and asymmetry of left and right side skeletal muscles were evaluated. Upper extremity skeletal muscles on the dominant side were significantly larger than those on the nondominant side in both Lenke1A and 5C groups. The asymmetry of upper extremity skeletal muscles was significantly greater in the Lenke1A group than in the Lenke5C group. Additionally, the size of the asymmetry did not correlate with the magnitude of the major curve and rotational deformation but did correlate with a right shoulder imbalance in the Lenke1A group. These results suggest that in AIS with a constructive thoracic curve, right shoulder imbalance is an independent risk factor for upper extremity skeletal muscle asymmetry.

## 1. Introduction

Asymmetry in limb muscle strength affects a variety of physical abilities. Lower-limb muscle power asymmetry has been reported to be related to gait speed and knee joint health [1,2,3]. Additionally, asymmetry of handgrip strength is associated with physical problems such as lower cognitive function, neurodegenerative disorders, functional disability, future falls, fractures, and shortened time to mortality in elderly people [4,5,6,7,8,9]. Therefore, further studies of asymmetry in limb muscles are required to develop well-targeted strategies for preventing mobility limitation in older people. However, the precise etiology of limb muscle asymmetry is still unknown, although trauma, sports, and stroke have been reported [1,10].

Adolescent idiopathic scoliosis (AIS) is a structural and lateral curvature with rotation of the spine that affects 1–3% of children and arises around puberty [11]. Various left-right imbalances, such as shoulder and body balance, have been observed in AIS. Many studies of AIS-related muscle asymmetry involve paravertebral, psoas, and masticatory muscles [12,13,14,15]. In contrast, asymmetry in limb muscles in patients with AIS remains unclear.

The purposes of the present study were to (1) quantify limb muscle asymmetry in patients with AIS; (2) compare AIS patients with major lumbar and major thoracic curves; (3) examine correlations between limb muscle asymmetry and radiographic parameters.

## 2. Materials and Methods

The study was approved for ethics by the institutional review board at the corresponding author’s institution. The study was approved for ethics by the institutional review board at the corresponding author’s institution (approved number 2556; the data of approval; 14 April 2018). Written informed consent was obtained from the parents or guardians of all subjects in the form. 

### 2.1. Patient Population

The medical records of 54 consecutive AIS patients, including those with sports experiences, were reviewed retrospectively. This study included eligible patients with AIS with major thoracolumbar/lumbar (TL/L) curves (Lenke type 5C) or MT curves (Lenke Type 1A) who underwent posterior spinal fusion surgery between July 2018 and August 2022 at our university hospitals. Patients with left-hand dominance were excluded.

### 2.2. Radiographic Parameters

Body composition was measured using whole-body dual-energy X-ray absorptiometry (DXA, QDR-DELPHIW scanner DPX-NT; Hologic, Waltham, MA, USA), which assessed lean muscle mass, soft tissue, fat, and whole-body bone minerals for all appendicular limb regions. Appendicular skeletal muscle mass was calculated as the sum of skeletal muscle mass in the arms and legs, assuming that the lean mass of muscle and soft tissue was representative of the skeletal muscle mass. The skeletal muscle mass index (SMI) was calculated as the sum of upper and lower limb soft tissue mass (kg/m^2^). The ratio of right to left upper arm skeletal muscle mass was defined as the asymmetry ratio (rt/lt). The difference in right and left upper arm skeletal muscle mass was defined as the asymmetry difference (rt-lt).

The Lenke classification defines a major T/TL/L curve with nonstructural thoracic curves (Cobb’s angle <25° on a side bending film). Standing whole spine posterior-anterior and lateral standing radiographs were evaluated by two surgeons (TO and GG). The magnitudes of the MT and TL/L curves were measured using Cobb’s method for the curve parameters. Additionally, clavicle angle (Cla-A) and radiographic shoulder height (RSH) were measured. Apical vertebral rotation was measured from preoperative computed tomography (CT). Radiographic measurements were obtained by two board-certified spine surgeons (Authors 1 and 2) to determine the interobserver error. The mean values of their measurements were used to calculate an intraclass coefficient of 0.893, indicating that the inter-rater reliability was almost ideal.

### 2.3. Statistical Analysis

Radiographic parameters were compared between the 1A and 5C groups. Mean ± SD values were reported for continuous variables, and number (percentage) values were used for categorical variables. Student’s *t*-tests, Mann-Whitney tests, or Fisher’s exact tests were used to compare the mean values between pre- and postsurgical patients, assuming normal distributions for continuous variables. We used Prism (version 9.0; GraphPad Software, La Jolla, CA, USA) to calculate summary statistics and perform *t*-tests. Asterisks indicate statistical significance (*p* < 0.05).

## 3. Results

### 3.1. Overall Data

There was no significant difference in age, gender, height, weight, body mass index (BMI), or SMI between the 1A and 5C groups. The mean (± standard error) Cobb angles of the MT and TL/L before surgery were 46.3° ± 8.8° and 39.3° ± 8.6°, respectively. Both Cla-A and RSH were significantly smaller in group 1A than in group 5C, meaning that group 1A had more cases of the right shoulder up.

### 3.2. Asymmetry in Skeletal Muscles of the Upper Arm in AIS Patients

In both 1A and 5C groups, forearm skeletal muscle mass was significantly greater on the right side, which was the dominant side (Figure 1A). Both the asymmetry ratio and the asymmetry difference of upper arm skeletal muscles were significantly larger in group 1A than in group 5C (Figure 1B). There was no significant difference in skeletal muscle mass between the left and right lower limbs in AIS patients (Figure 1C).

### 3.3. Sports

Based on their sporting experience, patients were divided into three groups: (1) no sports experience, (2) sports that primarily use the upper limb, (3) Other sports. There were no significant differences in the frequency of sports experience between the Lenke1A and 5C groups (Table 1). The asymmetry ratio and asymmetry difference of upper arm skeletal muscle mass in group (2) were significantly larger than in groups (1) and (3) (Figure 2).

### 3.4. Correlation between Asymmetry in Skeletal Muscles of the Upper Arm and Coronal Parameters in the Lenke1A Group

Cla-A and RSH were significantly correlated with both asymmetry ratio and asymmetry difference in skeletal muscles of the upper arm (Figure 3A,B). In contrast, the Cobb angle and AVR were not correlated in group 1A (Table 2). Figure 4 shows a representative radiograph of a Lenke1A AIS patient with the right shoulder up, who had a large asymmetry ratio of upper arm skeletal muscle mass. There were no significant correlations between asymmetry in skeletal muscles of the upper arm and coronal parameters (data not shown).

## 4. Discussion

The present study showed that upper extremity skeletal muscles on the dominant side were significantly larger than those on the nondominant side in both Lenke1A and 5C groups. Interestingly, we found that both the asymmetry ratio and asymmetry difference of upper extremity skeletal muscles were significantly greater in the Lenke1A group than in the Lenke5C group. Additionally, the size of the asymmetry did not correlate with the magnitude of the major curve or rotational deformation but did correlate with the right shoulder up in the Lenke1A group. These results suggest that in AIS with a constructive thoracic curve, right shoulder up is an independent risk factor for upper extremity skeletal muscle asymmetry. There are many reports focusing on muscle asymmetry in scoliosis involving paravertebral, psoas, and masticatory muscles [12,13,14]. However, to our knowledge, this is the first study to show asymmetry in upper extremity muscle mass in patients with AIS.

Currently, treatment for AIS focuses on the patient’s self-appearance, low back pain, and respiratory function. This study is the first to suggest that functional disabilities due to upper extremity skeletal muscle asymmetry in the elderly are a new target for AIS treatment. A recent study revealed that shoulder imbalance in AIS patients had a very large impact on their satisfaction with appearance [16]. Because postoperative shoulder imbalance (PSI) commonly causes dissatisfaction among AIS patients, numerous reports have focused on techniques and fixation ranges to reduce PSI [17,18,19,20]. Further studies are needed to clarify whether the asymmetry of the skeletal muscles of the upper extremity could be improved by correcting the shoulder balance through spinal corrective surgery.

One limitation of this study was that it did not identify the mechanism by which the right shoulder imbalance is involved in upper extremity skeletal muscle asymmetry. We believe that imaging analysis of the upper extremity skeletal muscle mass will be necessary in the future. A relationship has been demonstrated between bilateral strength imbalances and sports in which the dominant side is used more frequently, such as volleyball and basketball [21,22]. The association between sports experience and AIS also has been described [23]. The present study showed there might be a relationship between sports that primarily use the upper extremity and upper extremity skeletal muscle asymmetry. The possibility that the right shoulder up with a thoracic structural curve may result in the more frequent use of the right side compared to the left side should be considered in the future. This study suggests that even if 5C cases have shoulder balance, this can be compensated for in normal life by the nonstructural curve of the thoracic spine. In contrast, 1A cases with a right shoulder up due to a structural thoracic spine have a greater risk of left-right differences in upper limb use resulting in upper extremity skeletal muscle mass asymmetry.

## 5. Conclusions

The asymmetry ratio and asymmetry difference of upper extremity skeletal muscles were significantly greater in the Lenke1A group than in the Lenke5C group. The size of the asymmetry did not correlate with the magnitude of the major curve or rotational deformation but did correlate with the right shoulder up in the Lenke1A group.

## Figures and Tables

**Figure 1 jcm-11-07117-f001:**
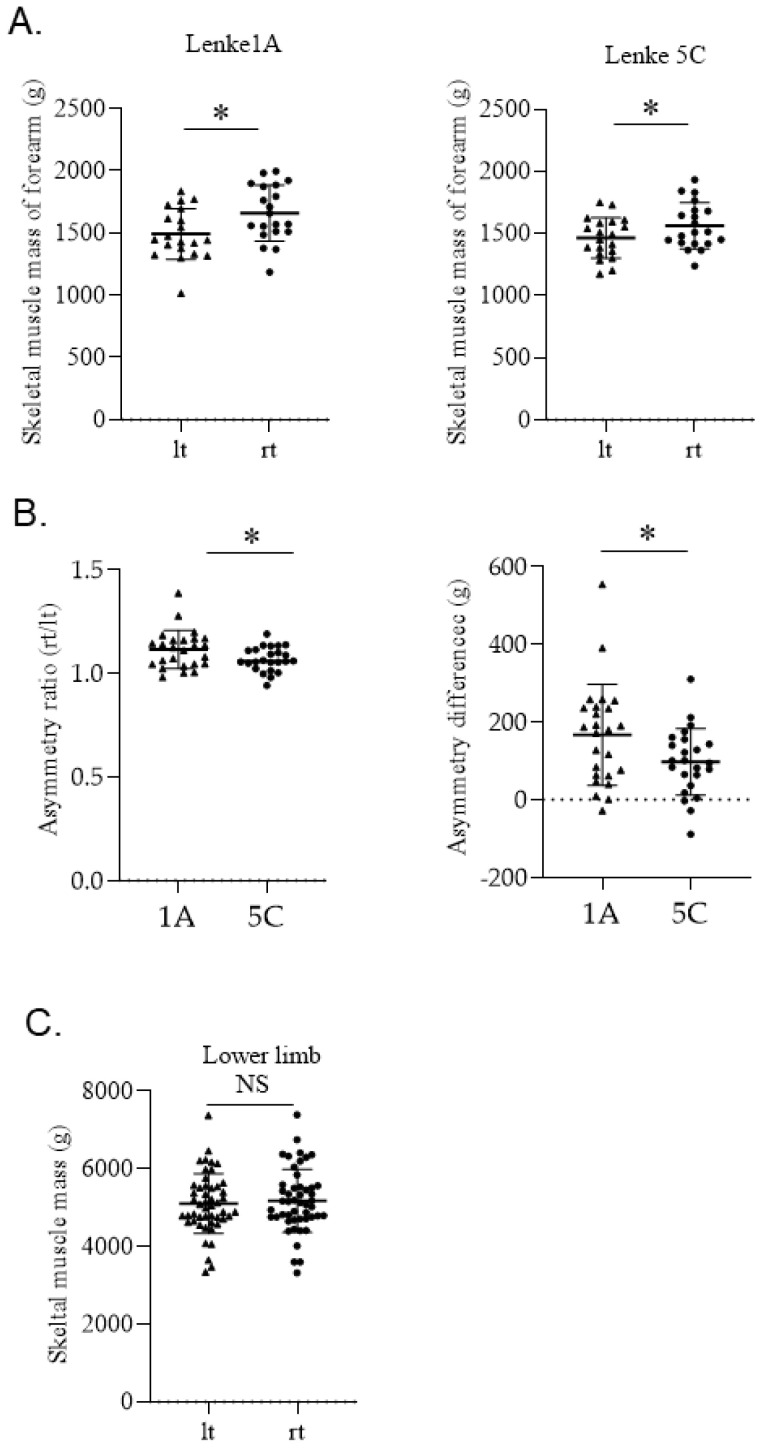
(**A**) Comparison of left and right skeletal muscles of the upper arm in Lenke1A or 5C groups. (* *p* < 0.005), (**B**) Comparison of asymmetry ratio (rt/lt) and asymmetry difference (rt−lt) for Lenke1A and 5C groups. (* *p* < 0.005), (**C**) Comparison of left and right skeletal muscles of the lower limb in all AIS cases. (NS; not significant).

**Figure 2 jcm-11-07117-f002:**
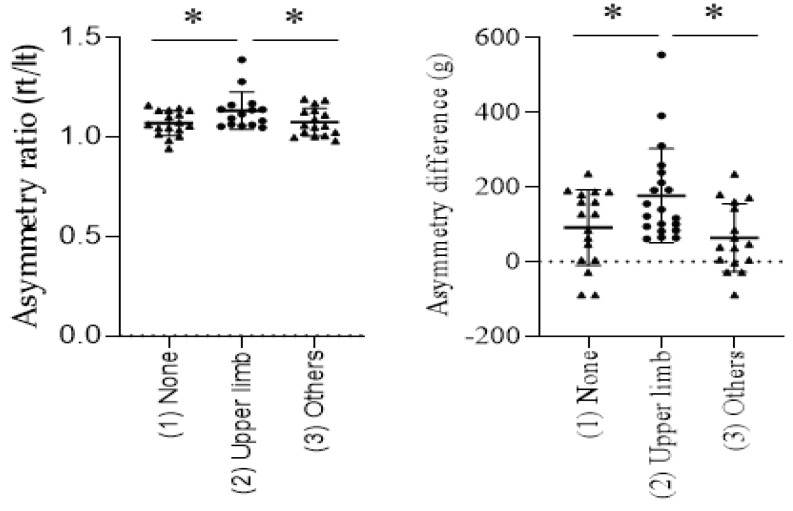
Comparison of asymmetry ratio (rt/lt) and asymmetry difference for (1) No sports experience, (2) Sports that primarily use the upper limb, and (3) Other sports in all AIS cases. (* *p* < 0.005).

**Figure 3 jcm-11-07117-f003:**
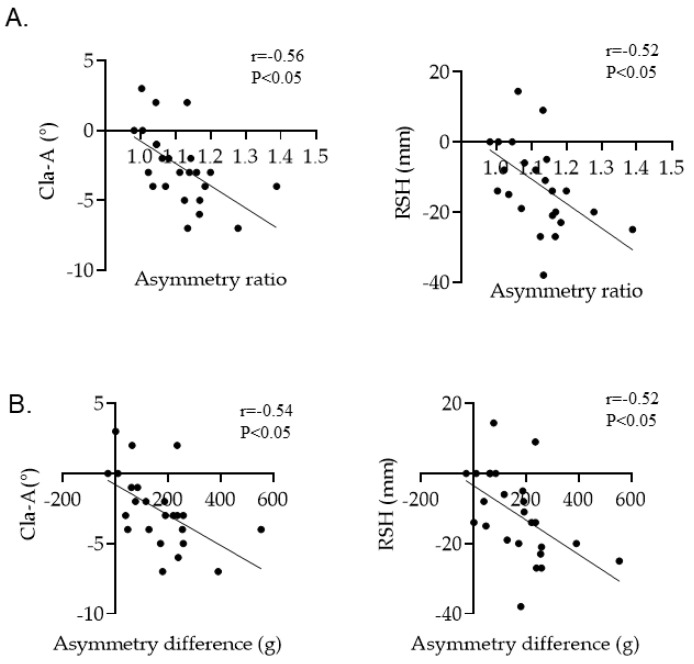
(**A**) Correlation between Cla-A, RSH, and asymmetry ratio in the Lenke1A group, (**B**) Correlation between Cla-A, RSH, and asymmetry difference in the Lenke1A group.

**Figure 4 jcm-11-07117-f004:**
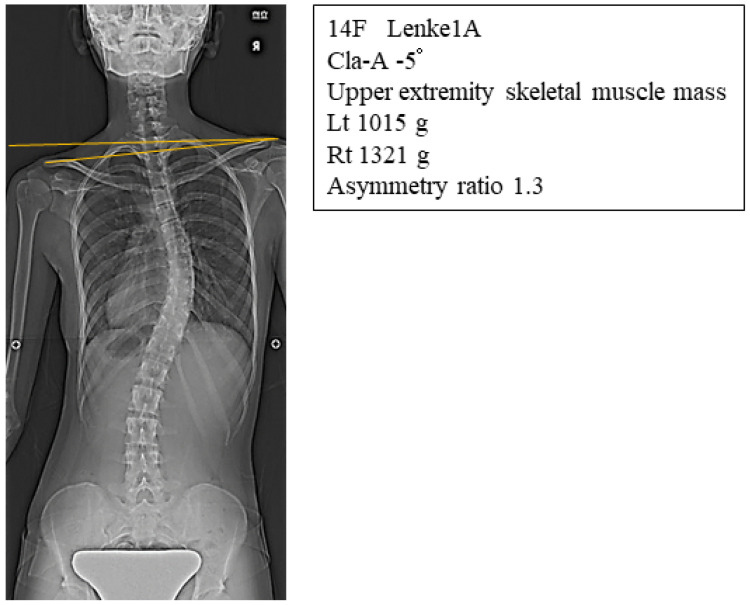
Representative radiograph of a Lenke1A AIS patient with a right shoulder imbalance who had a large asymmetry ratio of upper arm skeletal muscle mass.

**Table 1 jcm-11-07117-t001:** Preoperative patient characteristics, radiographic measurements.

Variable	AIS (*n* = 54)		
Lenke Classification	1A (*n* = 28)	5C (*n* = 26)	*p* Values
Age (y)	16.2 ± 2.5	16.0 ± 2.7	NS
Female/male gender (*n*)	26/2	26/0	NS
Risser grade (*n*)			NS
1	1	1	
2	1	2	
3	0	4	
4	13	9	
5	13	10	
Height (cm)	159.7 ± 4.0	158.9 ± 5.6	NS
Weight (kg)	48.9 ± 4.7	48.3 ± 5.3	NS
BMI (kg/m^2^)	19.2 ± 1.9	19.1 ± 2.0	NS
SMI (kg/m^2^)	5.2 ± 0.8	5.4 ± 0.5	NS
Upper arm skeletal muscles			
Asymmetry difference (g)	166.7 ± 130	97.6 ± 85.6	0.034
Asymmetry ratio	1.12 ± 0.09	1.07 ± 0.06	0.037
Sports experiences			
None	8	12	
Sports that primarily use the upper limb	10	7	
Tennis	2	3	
Table tennis	2	0	
Volleyball	2	1	
Basketball	1	2	
Softball	3	0	
Badminton	0	1	
Other sports	10	7	
Classical ballet	0	1	
Dancing	1	2	
Athletics	3	0	
Soccer	0	2	
Combat sport	5	1	
Swimming	2	1	
Coronal parameters			
MT cobb angles (°)	46.3 ± 8.8		
TL/L cobb angles (°)		39.3 ± 8.6	
Cla-A (°)	−1.4 ± 2.8	0.48 ± 1.8	*p* < 0.05
RSH (mm)	−10.5 ± 13.1	0.93 ± 10.9	*p* < 0.05

Body mass index (BMI); Skeletal muscle mass index (SMI); Clavicle angle (Cla-A); No significant difference (NS); Major Thoracic (MT); Radiographic shoulder height (RSH).

**Table 2 jcm-11-07117-t002:** Correlations between asymmetry in skeletal muscles of the upper arm and coronal parameters in group 1A.

		Asymmetry Ratio	Asymmetry Difference
Cla-A (°)	R	−0.56	−0.54
	*p*	<0.05 *	<0.05 *
RSH (mm)	R	−0.52	−0.52
	*p*	<0.05 *	<0.05 *
Cobb angles (°)	R		
	*p*	NS	NS
AVR (°)	R		
	*p*	NS	NS

Clavicle angle (Cla-A); Radiographic shoulder height (RSH); Apical vertebral rotation (AVR); There is a significant difference (*); No significant difference (NS); Correlation coefficient (R).

## Data Availability

The data presented in this study are available on request from the corresponding author.

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
