# Peer review of "Upper Extremity Skeletal Muscle Mass Asymmetry Exacerbated by Shoulder Imbalance in Lenke1A Adolescent Idiopathic Scoliosis"

_jcm, 2022, doi:10.3390/jcm11237117_

Round 1

Reviewer 1 Report

The authors explored the relationship between the Upper extremity skeletal muscle mass asymmetry and adolescent idiopathic scoliosis, which was a novel idea. However, there were some points should be clarified or improved.

1) I think the study design at the Line-169 should be considered, because the study was conducted through a retrospective review of patients, and the study belongs to a cross-sectional study.

2) Sence the study was a correlation analysis between the upper extremity muscle mass and AIS, as well as a cross-sectional study, I don't think the conclusion is suitable for the expression which includes the risk factors indicating a causal relationship.

3) It would be better that the primary outcomes related muscle mass are illuminated by images.

4) A conclusion section should be added behind the discussion section to make the findings more highlighting conspicuous.

Author Response

Comments and Suggestions for Authors

The authors explored the relationship between the Upper extremity skeletal muscle mass asymmetry and adolescent idiopathic scoliosis, which was a novel idea. However, there were some points should be clarified or improved.

1) I think the study design at the Line-169 should be considered, because the study was conducted through a retrospective review of patients, and the study belongs to a cross-sectional study.

→Study design has been revised to " a cross-sectional study " as this suggestion.

2) Sence the study was a correlation analysis between the upper extremity muscle mass and AIS, as well as a cross-sectional study, I don't think the conclusion is suitable for the expression which includes the risk factors indicating a causal relationship.

→According to this suggestion, we have added a conclusion section behind the discussion section on what we found from the results of present study (Line173-178).

3) It would be better that the primary outcomes related muscle mass are illuminated by images.

→We agree with this suggestion. We could not examine the upper extremity in present study because it was not imaged, but we believe that imaging analysis will be necessary in the future. We added this point as a limitation (Line159-161).

4) A conclusion section should be added behind the discussion section to make the findings more highlighting conspicuous.

→According to this suggestion, we have added a conclusion section behind the discussion section on what we found from the results of present study (Line173-178).

Reviewer 2 Report

Dear editor, thank you for giving me the chance to evaluate this work. I have read the article carefully and I would like to make a few suggestions, although it is considered sufficient in general.

It was repeatedly reported that muscle volume is higher on the dominant side in normal individuals. Therefore, it is not surprising that this difference is also observed in scoliosis patients. The authors mentioned that they found an interesting difference regarding asymmetry ratio and asymmetry difference of upper extremity skeletal muscles. However, when the results section is examined, this difference has not been shown successfully enough. It would be appropriate to add a table showing the difference between the groups on the subject.

It is also striking that the authors did not adequately discuss their results with the literature. There are not few studies on scoliosis and muscle mass in the literature. It would be appropriate for the authors to discuss their findings about upper extremity muscle mass, which they also considered in the title, with the literature.

Author Response

Dear editor, thank you for giving me the chance to evaluate this work. I have read the article carefully and I would like to make a few suggestions, although it is considered sufficient in general.

It was repeatedly reported that muscle volume is higher on the dominant side in normal individuals. Therefore, it is not surprising that this difference is also observed in scoliosis patients. The authors mentioned that they found an interesting difference regarding asymmetry ratio and asymmetry difference of upper extremity skeletal muscles. However, when the results section is examined, this difference has not been shown successfully enough. It would be appropriate to add a table showing the difference between the groups on the subject.

→We agree with this suggestion. An important point of present study is that the “magnitude of asymmetry” was greater in the thoracic curve group (Lenke1A) than in the lumbar curve group (Lenke5C). This result was represented in the graph in Figure 1, we have added this results in the Table1.

It is also striking that the authors did not adequately discuss their results with the literature. There are not few studies on scoliosis and muscle mass in the literature. It would be appropriate for the authors to discuss their findings about upper extremity muscle mass, which they also considered in the title, with the literature.

→We appreciate this suggestion to further improve our discussion. As this suggestion, there have been many reports of asymmetries in trunk muscle and masticatory muscles in scoliosis, but I believe this study is the first to report about asymmetry of upper limb muscle strength. Now this point was presented in discussion section (Line145-147).

Round 2

Reviewer 1 Report

The manuscript was improved after revision!

Author Response

The manuscript was improved after revision!

→Thank you for your great review.